

# Downscaling GCM data for climate change impact assessments on rainfall: a practical application for the Brahmani-Baitarani river basin

R. J. Dahm[1], U. K. Singh[2], M. Lal[2], M. Marchand[1], F. C. Sperna Weiland[1], S. K. Singh[3], and M. P. Singh[3]

[1]Deltares, Boussinesqweg 1, P.O. Box 177, 2600 MH Delft, the Netherlands
[2]RMSI, A-8 Sector 16, NOIDA 201 30, India
[3]Central Water Commission, R.K. Puram, New Delhi 110 066, India

Received: 16 November 2015 – Accepted: 10 December 2015 – Published: 15 January 2016

Correspondence to: R. J. Dahm (ruben.dahm@deltares.nl)

Published by Copernicus Publications on behalf of the European Geosciences Union.

HESSD

doi:10.5194/hess-2015-499

Downscaling GCM data for climate change impact assessments on rainfall

R. J. Dahm et al.

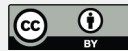

## Abstract

The delta of the Brahmani-Baitarani river basin, located in the eastern part of India, frequently experiences severe floods. For flood risk analysis and water system design, insights in the possible future changes in extreme rainfall events caused by climate change are of major importance. There is a wide range of statistical and dynamical downscaling and bias-correction methods available to generate local climate projections that also consider changes in rainfall extremes. Yet the applicability of these methods highly depends on availability of meteorological observations at local level. In the developing countries data and model availability may be limited, either due to the lack of actual existence of these data or because political data sensitivity hampers open sharing.

We here present the climate change analysis we performed for the Brahmani-Baitarani river basin focusing on changes in four selected indices for rainfall extremes using data from three performance-based selected GCMs that are part of the 5th Coupled Model Intercomparison Project (CMIP5). We apply and compare two widely used and easy to implement bias correction approaches. These methods were selected as best suited due to the absence of reliable long historic meteorological data. We present the main changes – likely increases in monsoon rainfall especially in the Mountainous regions and a likely increase of the number of heavy rain days. In addition, we discuss the gap between state-of-the-art downscaling techniques and the actual options one is faced with in local scale climate change assessments.

## 1 Introduction

In the past extreme rainfall over India has resulted in landslides, flash floods, severe river floods, and crop damage that have had major impacts on society, the economy, and the environment (Goswami et al., 2006). Increases in rainfall extremes have quantifiable impacts on intensity–duration–frequency relations (Kao and Ganguly, 2011) and

Discussion Paper | Discussion Paper | Discussion Paper | Discussion Paper |

# HESSD

doi:10.5194/hess-2015-499

**Downscaling GCM data for climate change impact assessments on rainfall**

R. J. Dahm et al.

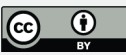

are expected to enhance coastal and river flood risks and will, without adaptive measures, substantially increase flood damages. Climate adaptation strategies for emergency planning, the design of engineering structures, reservoir management, pollution control, risk calculations rely on knowledge of the frequency of these extreme rainfall

events (Guhathakurta et al., 2011; Goswami et al., 2006).

Several studies already investigated the trends in rainfall extremes over India (Parthasarathy et al., 1993; Dash et al., 2009; Ramesh Kumar et al., 2009; Chaturvedi et al., 2012 and many others). Goswami et al. (2006) found significant positive trends in the frequency and the magnitude of heavy rain events and a significant negative

trend in the frequency of moderate events over central India during the monsoon seasons from 1951 to 2000. Climate change is expected to further alter the intensity and frequency of extreme rainfalls (Chatuverdi et al., 2012; Frei et al., 2006; Kharin et al., 2007). Globally the intensity of extreme rainfall is projected to increase even in regions where mean rainfall decreases (Semenov and Bengtsson, 2002). Moreover, future cli-

mate studies based on climate model simulations suggest that greenhouse warming is likely to intensify the monsoon rainfall over a broad region encompassing South Asia (Lal et al., 2000; May, 2011; Meehl and Arblaster, 2003; Rupakumar et al., 2006; Trenberth, 2011). However, precise assessments of future changes in the regional monsoon rainfall have remained ambiguous due to wide variations among the model projections

(Kripalani et al., 2007; Sabade et al., 2011; Turner and Annamalai, 2012). And the simulated rainfall response to global warming by climate models is actually accompanied by a weakening of the large-scale South-West (SW) monsoon flow (Kripalani et al., 2003; Krishnan et al., 2013; Ueda et al., 2006). Nonetheless, the recently released CMIP5 projections confirm significant reduction in return times of annual extremes of

daily rainfall for the late 21st century in India (Ramesh and Goswami, 2014) and increases in seasonal mean rainfall amounts (Menon et al., 2013). In addition, Chaturvedi et al. (2012) found a steady increase in the number of days with extreme rainfall for the period of 2060 and beyond.

**HESSD**

doi:10.5194/hess-2015-499

**Downscaling GCM data for climate change impact assessments on rainfall**

R. J. Dahm et al.

Within this paper we try to assess future climate induced changes in several rainfall indices of relevance to water resources and flood risk management in the Brahmani-Baitarani river basin. The basin is located in the eastern part of India. It experienced severe floods in recent years of 2001, 2003, 2006, 2008, 2011 and 2013 (Government of Odisha, 2011; Government of Odisha, 2013).

To enable a quantitative climate change analysis we use Global Climate Model (GCM) data. GCM rainfall data normally have biases from observations and needs to be corrected to ensure its applicability at the local scale (Teutschbein and Seibert, 2012; Sperna Weiland et al., 2012; Christensen et al., 2008). There is a wide range of statistical and dynamical downscaling and bias-correction methods available to generate local climate projections that also consider changes in rainfall extremes. Yet the applicability of these methods highly depends on availability of meteorological observations at local scale. In this study, data from a limited number of rain gauges with incomplete time-series and a lower resolution gridded rainfall product – the APHRODITE dataset (Yatagai et al., 2012) – could be accessed and used. Due to this data quality issue we concentrate on two widely used and easy to implement bias correction procedures – Linear Scaling (LS) and Delta Change (DC). We evaluate the influence of these procedures on resulting changes in rainfall indices. Instead of using a large ensemble of GCMs we focus on three models that nearly span the full range of annual mean rainfall projections available for India – analysing in more detail the locally relevant biases within these three GCMs and the changes they project.

This paper is organized as follows. Section 2 presents an overview of the selected GCMs, reference dataset, and bias correction methods. This is followed by a short description of the impact analysis framework applied and the study area. In Sect. 3 we report the main findings of our study and present the range of rainfall indices under climate change. Here, we are specifically concerned with an inter- and intra-comparison of the GCMs and downscaling methods. Next the limitations experienced for bias-correction techniques experienced in this study are discussed. Section 5 summarizes

our main conclusions and provide outlook for future work and practical applications in the study area.

## 2 Methods

The method adopted in this study for the analysis of climate change impact on four rainfall indices is based on four steps described in subsequent sub-sections.

1. GCM selection by comparing observed (gauge and gridded time series) and simulated (GCM) variability of monthly rainfall.

2. A description of the selected reference rainfall dataset.

3. Bias correction of daily rainfall time series from the selected GCMs for the baseline and future projections using Linear Scaling and Delta Change method.

4. Rainfall indices analysis for the baseline and future projections focussing on water resources and flood risk management in Brahmani-Baitarani river basin.

### 2.1 Global Circulation Model selection

Data sets generated in simulations from three GCMs undertaken for the Fifth Coupled Model Inter-comparison Project (CMIP5 – Taylor et al., 2012; Knutti and Sedláček, 2013) have been downloaded which are the Hadley Center Global Environment Model 2 – Earth System (HadGEM2-ES), the GFDL-CM3, and the MIROC-ESM (see Table 1). Data for the time slices representing periods 1961–1990 (baseline), short term climate change projection (2030–2059), and long term projection (2060–2099) were used.

Almost all CMIP5 models show good performance for surface air temperature simulations averaged over South Asia and the Indian Sub-continent, with MIROC5 as one of the models closest to the observations. Yet for area-averaged annual and seasonal rainfall the models significantly deviate from observations over India (Chaturvedi et al.,

Discussion Paper | Discussion Paper | Discussion Paper | Discussion Paper

**HESSD**

doi:10.5194/hess-2015-499

**Downscaling GCM data for climate change impact assessments on rainfall**

R. J. Dahm et al.

**HESSD**

doi:10.5194/hess-2015-499

**Downscaling GCM data for climate change impact assessments on rainfall**

R. J. Dahm et al.

2012). Here GFDL-CM3 is one of the best performing models. In addition, HadGEM2-ES is selected because it is one of the most commonly used GCMs. For India the three selected GCMs nearly span the uncertainty band for annual mean rainfall that was obtained with 18 Earth System Models (ESMs) used in a validation exercise (Mondal and
Mujumdar, 2014; Rameshkumar and Goswami, 2014). Interpreting Menon et al. (2013), HadGEM2-ES could be regarded as "too dry", GFDL-CM3 as "representative", and MIROC-ESM as "quite wet".

## 2.2  Representative Concentration Pathway (RCP) Selection

The RCPs as considered in IPCC AR5 (Vuuren et al., 2011) are compatible with the
full range of stabilization, mitigation, and baseline emission scenarios, and span a full range of socio-economic driving forces (Hibbard et al., 2011). In this study, we considered RCP6.0 for the generation of the climate change projections as it follows a stabilizing $CO_2$ concentration close to the median range of all four policy pathways and will give us a not too extreme indication of what might change. Most likely the spread and
uncertainty in projections will increase when including additional RCPs and changes may become more pronounced when using the extreme RCP8.5 (Hibbard et al., 2011).

## 2.3  Observed rainfall

Observed rainfall was obtained from the APHRODITE long-term daily gridded precipitation dataset (version V1101) for Monsoon Asia (Yatagai et al., 2012). The APHRODITE
(Asian Precipitation Highly Resolved Observational Data Integration Towards Evaluation of Water Resources, Japan) is a long-term (1951–2007) continental-scale product that contains a dense network of daily rain gauge data for Asia including the Himalayas, South and Southeast Asia. The APHRODITE dataset consists of 0.25° × 0.25° resolution gridded daily precipitation derived from the Global Telecommunication System
(GTS), precompiled datasets and APHRODITE's individual data collection. It covers the period 1951–2007. The data is quality controlled and corrected for orographic ef-

fects. It includes over 2000 stations over India and captures the large scale features of monsoon rainfall over the Indian region well (Rajeevan and Bhate, 2008). Hereafter it is referred to as the climatological rainfall dataset. Within this paper we perform additional validation focusing on extremes using observations from three rain gauges that were obtained from the Central Water Commission of India (CWC). Locations of the rain gauges are listed in Table 2 and shown in Fig. 1. Gauge observations could not be used as reference since they do not overlap with the full GCM baseline period.

## 2.4  Bias correction

There is often a clear bias from observations in the statistics of variables produced by GCMs such as temperature and rainfall due to limitations in among others the incorporation of local topography and non-stationary phenomena within the GCMs. GCM outputs can therefore often not be directly applied for impact studies at the catchment scale (Christensen et al., 2008; Kay et al., 2006; Kotlarski, 2005; Fan et al., 2010). Dynamic downscaling, statistical downscaling, and bias correction are the most commonly used methods to generate locally applicable climate data (Bergstrom, 2001; Fowler et al., 2007; Schoof et al., 2009). Dynamic downscaling includes nesting of high resolution Regional Climate Models (RCMs) with GCM outputs at boundaries which ensures consistency between climatological variables. However, they are computationally expensive, their skill strongly depends on GCM boundary and for India there are no ready to use RCM scenarios for the RCP emission scenarios at hand. Statistical downscaling models, on the other hand, are based on statistical relationships between large-scale climate variables (predictors) and local-scale climate variables (predictant) and hence require less computational time. Yet for the establishment of reliable statistical relationships long historical observed records of both predictor and predictant variables should be available.

Simpler bias correction procedures consisting of general transformation techniques for adjusting the GCM output time series are often used. They assume a stationary bias between GCM output and observations for the baseline period and future climate

**HESSD**

doi:10.5194/hess-2015-499

Downscaling GCM data for climate change impact assessments on rainfall

R. J. Dahm et al.

(Teutschbein and Seibert, 2012a). With the more advanced Quantile Mapping method (Thrasher et al., 2012; Camici et al; 2014) changes in mean and extreme rainfall are corrected individually – yet this method requires reliable CDFs of observed rainfall and from the comparison between the APHRODITE dataset and station observations (Rana et al., 2014) we know that large biases in rainfall extremes exist.

Therefore, we restrict ourselves to two simple methods (1) the Delta Change method and (2) the Linear Scaling method and applied these to the data extracted for our region of interest from all three GCM outputs (Christensen et al., 2008; Chen et al., 2012), realizing that these methods only concentrate on the correction of monthly mean rainfall amounts and can thus affect our analysis of changes in extreme rainfall indices.

### 2.4.1 Delta change method

The Delta Change method transforms historical observations into future projections using monthly average correction factors that are derived from the GCM simulations for the baseline and future climate (Camici et al., 2014; Chen et al., 2013; Teutschbein et al., 2012b) according to:

$$\Delta P_m = \frac{\overline{P\_GCM_m^{fut}}}{\overline{P\_GCM_m^{BL}}} \tag{1}$$

$$P\_OBS^{fut} = \Delta P_m P\_OBS^{BL} \tag{2}$$

where fut refers to future climate, BL refers to baseline climate (i.e., 1961–1990), GCM refers to GCM simulations and OBS refers to either observations (BL period) or future projections (fut), $P$ refers to daily rainfall values and $P_m$ refers to monthly long term mean rainfall values.

The disadvantage of this method is that the future and baseline scenarios differ only in terms of their means and intensity while all other statistics of the data, such as the skewness and number of wet days, remain almost unchanged (Camici et al., 2014).

**HESSD**

doi:10.5194/hess-2015-499

**Downscaling GCM data for climate change impact assessments on rainfall**

R. J. Dahm et al.

This will hamper the assessment of change in extreme rainfall frequency as the change in return period is essentially a derivative of the multiple factors.

### 2.4.2 Linear scaling method

Within the Linear Scaling method the long-term average monthly bias between the baseline (i.e., 1961–1990) GCM simulations and observations is derived and this bias is applied to correct the future GCM simulations assuming a stationary bias over time, according to:

$$BIAS_m = \frac{\overline{P\_OBS_m^{BL}}}{\overline{P\_GCM_m^{BL}}} \tag{3}$$

$$P\_OBS^{fut} = BIAS_m P\_GCM^{fut} \tag{4}$$

where $BIAS_m$ is the monthly average bias for the baseline climate and the correction factor for the future GCM simulations. The advantage of this method is that the variability of the corrected data, for both baseline and future climate is more consistent with the GCM data, which implies that changes in wet day frequencies and intensities can be derived, yet they are not individually corrected as all events are adjusted with the same monthly average correction factor (Teutschbein et al., 2012b).

### 2.5 Impact analysis framework

To study the impact of climate change on rainfall and associated fields like water resources and flood risk management, we selected the following four indices.

1. Mean annual and seasonal rainfall in order to explore possible effects of climate change on water resources and flood risk management.

Discussion Paper | Discussion Paper | Discussion Paper | Discussion Paper |

**HESSD**

doi:10.5194/hess-2015-499

**Downscaling GCM data for climate change impact assessments on rainfall**

R. J. Dahm et al.

2. Frequency of seasonal Wet Days (WD). According to the definition followed by the India Meteorological Department (IMD), a day with rainfall is considered a "wet day" if the rainfall equals or exceeds 2.5 mm.

3. Seasonal frequency of days with light, moderate, and heavy rainfall (LRD, MRD, and HRD respectively). Table 3 shows the IMD classification for rainfall intensities. We clustered all "heavy rain" classifications into one group to examine climate change induced alterations in potential flood inducing events.

4. One-day extreme rainfall return periods to investigate the potential impact of climate change on rainfall intensities. In this study we applied the Gumbel extreme value type-1 distribution (Gumbel, 1941) to derive rainfall intensities for different return periods (2, 5, 10, 25, 30, 50, and 100 years).

Table 4 shows the classification of seasons according to IMD.

## 2.6 Study area

This study focusses on the Brahmani-Baitarani river basin located in the eastern part of India. The Brahmani-Baitarani river basin and neighbouring Mahanadi river basin experienced serious floods in the years 2001, 2003, 2006, 2008, 2011 and 2013. During September 2011 heavy rainfall together with high sea levels led to the flooding of large parts of the Delta. It affected about 3.4 million people of which 45 people lost their life (Government of Odisha, 2011). In 2013 cyclone Phailin created havoc along the coastal districts of Odisha state. Due to storm surge up to 3.5 m, large areas were inundated. The Baitarani River, along with other rivers, experienced floods as a result of torrential downpour. No less than 13 million people were affected of which 44 people lost their life (Government of Odisha, 2013).

The basin is an inter-state basin and spreads across the states of Chhattisgarh, Jharkhand, and Odisha. The elevation ranges from > 750 m in the north-western part of the basin to approximately 10 m in the delta region. The Baitarani River enters the

**HESSD**

doi:10.5194/hess-2015-499

**Downscaling GCM data for climate change impact assessments on rainfall**

R. J. Dahm et al.

**HESSD**

doi:10.5194/hess-2015-499

**Downscaling GCM data for climate change impact assessments on rainfall**

R. J. Dahm et al.

Brahmani River in the deltaic region before it drains into the Bay of Bengal. The catchment area is 51 822 km². The region is characterized by a sub-tropical monsoon climate zone with mean annual rainfall of approximately 1450 mm (CWC, 2011) most of which occurs during the SW monsoon season (June to September).

⁵ The basin is exposed to orographic effects with a positive elevation-rainfall depth slope during monsoon season and an inverse slope in the post-monsoon season when cyclones hit the coast of Odisha (Deltares, 2015). Due to the size and shape, the Brahmani-Baitarani river basin is captured by approximately one GCM grid cell in the deltaic region and one grid cell in the Mountainous region. This allows studying possible ¹⁰ orographic effects present in the future projections (Prokop and Walanus, 2013). GCM and climatological observed rainfall values were taken from the grid cell with center coordinates closest to these regions. It was decided not to resample the climatological observed and GCM datasets to one common grid, since up-scaling or spatial-averaging would result in spatial smoothening of rainfall extremes. Figure 2 shows the locations ¹⁵ of the grid cells used. Table 5 presents elevations of center grid points for both GCMs and observed climatological data derived from the SRTM 30.

## 3 Results

We will focus our discussion of the results on the SW monsoon season only as mainly changes in monsoon rainfall will affect water resources and flood risk management in ²⁰ the Brahmani-Baitarani river basin. The figures contain results for all seasons.

### 3.1 Comparison of climatological and gauge rainfall data

The climatological dataset covers the full GCM baseline period whereas the gauge observations are more recent and only available for a limited time-spam, they can therefore not be used as baseline reference. We here verify the quality of the climatological ²⁵ dataset compared to gauge observations for the overlapping period 1990–2007. Over-

**HESSD**

doi:10.5194/hess-2015-499

**Downscaling GCM data for climate change impact assessments on rainfall**

R. J. Dahm et al.

all the climatological dataset resembles the CWC rain gauges quite well (see Fig. 3). Yet during pre-monsoon and monsoon season climatological data underestimates the rainfall in comparison to the gauge observations (averaged for the three gauge stations 17 and 12 % less rain during the pre-monsoon and monsoon period respectively).

⁵ Climatological data underestimates the occurrence of "heavy rain" events during the 1990–2007 period with 68 % (24 instead of 74 events), 61 % (17 instead of 44 events), and 90 % (5 instead of 48 events) for Akhuapada, Anandpur, and Tilga gauge respectively, with an average 0.3–1.3 of such events per year (a frequency of less than 1 and changes therein will affect the analysis of extreme value return periods). This underes-

¹⁰ timation is most likely caused by the smoothening introduced when interpolating point observations to the grid. Differences in rainfall extremes between the two datasets can clearly be seen from their respective CDFs (see Fig. 4). Still, based on (1) this analysis, (2) the period covered by the climatological dataset and (3) the work of Rajeevan and Bhate (2008) we conclude that the dataset is suitable as a gridded reference dataset

¹⁵ for the Brahmani-Baitarani river basin. When quantifying the impact of climate change to strategic flood management studies in practice, the rainfall deviations between climatological and gauge extremes should be considered.

## 3.2  Comparison of climatological and GCM baseline data

The South-West (SW) monsoon, lasting from June to September, is the most impor-
²⁰ tant feature of the Indian climate. Its onset and withdrawal dates are highly variable. Over the period 2000–2012 the actual onset date over Kerala coast varied from the 23 May to 8 June (Puranik et al., 2013). As per the IMD the normal monsoon onset date for the Brahmani-Baitarani river basin is 10 June. However, during 2005–2012, the actual onset dates varied from 6 to 26 June (Indian Meteorological Department

²⁵ n.d. (a)). As a consequence, total rainfall in the month of June varies significantly in the Brahmani-Baitarani river basin. The baseline simulations of the three GCMs show a fair comparison with the observed rainfall on annual basis (see Fig. 5). However, for the monsoon season, the baseline simulations show an underestimation of average



seasonal rainfall. Rainfall amount in June is underestimated by both HadGEM2-ES and MIROC-ESM (11 and 19 % respectively of observed), and June under-estimations are especially large for the deltaic region where the GCM cells cover both land and sea.

Monthly rainfall amounts of HadGEM2-ES are almost all below climatological values, this corresponds with findings of Menon et al. (2013) that HadGEM2-ES is "too dry" compared to long-year observations. Their conclusion that MIROC-ESM is "quite wet" does not hold throughout the year for the baseline period for the Brahmani-Baitarani river basin. MIROC-ESM underestimates rainfall in February–May in the delta region and during the monsoon in the Mountainous region. GFDL-CM3 overestimates rainfall during the monsoon season in the deltaic region but overall performs best.

## 3.3 Impact of downscaling method on future GCM projections

We made a modification to the LS method for the month June as the underestimation of monsoon rainfall in June by HadGEM2-ES and MIROC-ESM would result in extreme large correction factors, unrealistic rainfall amounts (up to 900 mm day$^{-1}$) and consequently an unrealistic large increase in heavy rain events occurring in June when applying the LS method, see Table 6. Therefore, we decided to apply the July multiplier also for June. By doing so, the GCM total monsoon rainfall amount of MIROC-ESM and HadGEM2-ES become closer to the climatological dataset. Only for GFDL-CM3 total monsoon rainfall becomes slightly lower than that of climatological values. By applying the multiplier of July in June the annual mean baseline rainfall amounts will be somewhat different between the LS and DC method. Although the underestimation of rainfall by MIROC-ESM for the winter season in the Delta region leads to high multipliers as well, we decided not to apply a tailored correction here as it would only be applicable for one of the GCMs.

## HESSD

doi:10.5194/hess-2015-499

**Downscaling GCM data for climate change impact assessments on rainfall**

R. J. Dahm et al.

### 3.3.1 Mean seasonal rainfall

Figure 6 presents the mean seasonal rainfall for baseline and future climate. Nine panels are used to summarize our findings, showing for each GCM individually changes for the Delta region (left), the Mountainous region (middle) and for the basin as a whole (right). Towards the end of the century, monsoon rainfall is projected to increase throughout the basin with the Delta region in MIROC-ESM as only exception. For the shorter time horizon the signal is more variable, according to GFDL-CM3, monsoon rainfall will first decrease, whereas the other regions project increases except for the Delta region in MIROC-ESM. Both HadGEM2-ES and GFDL-CM3 also project increases for the pre-monsoon season. Increases are in general larger for the DC method; the multipliers of the LS method may reduce the change signal. These projections are in line with the study of Guhathakurta et al. (2011), Lal et al. (2000), May (2011), Rupakumar et al. (2006) and Menon et al. (2013) who also reported increases for monsoon rainfall in east and north east India.

### 3.3.2 Number of wet days

The projected change in the number of wet days (WD – Fig. 7) highly depends on the correction method used and the direction of the bias in the GCM data. For the DC method, the impact of climate change is given by the difference between the future GCM and the climatological data to which the DC method was applied. For the LS method there is a mismatch in the number of WD for the GCM data for the baseline period and the climatological data. The impact of climate change can thus only be derived when comparing future GCM data with the bias correction baseline when using the LS method.

Changes in this index cannot be captured well by the DC method. With the DC method the observed day-to-day variability remains unchanged in the future time-series and intensities are only scaled with monthly average correction factors (Teutschbein et al., 2012a; Camici et al., 2014). Changes are therefore directly related to the sea-

# HESSD

doi:10.5194/hess-2015-499

## Downscaling GCM data for climate change impact assessments on rainfall

R. J. Dahm et al.

Interactive Discussion

Discussion Paper | Discussion Paper | Discussion Paper | Discussion Paper | Discussion Paper |

sonal mean rainfall and only slightly differ due to the definition of the rainfall amount for a wet day. To project changes in the WD, while applying LS as correction method, the number of wet days has already been captured well by the GCMs for the future time period. Both HadGEM2-ES and MIROC-ESM show a distinct increase in WD in both regions and for both short and long term projections. While the "too dry" and "too wet" GCMs (Menon et al., 2013) show a clear trend, the trend in the "representative" GCM (GFDL-CM3) is minimal from a small decrease for the short term projections to a small increase in WD in the long term projections. From the mixed signals presented in Fig. 8, no clear conclusions on the potential direction of change in the WD during the monsoon season can be drawn, on average there is an indication for an increase. This increase was also found by Chaturvedi et al. (2012).

For the LS method we conclude that for a reliable projection of change in WD, the WD in the GCM baseline period should be close to observed, otherwise the correction method and bias from observations will disturb the change signal too much. For GFDL-CM3 we have seen that the number of WD during the baseline period is 103, which already nearly corresponds to the entire Monsoon season which explains the small increase found for this GCM. With the DC method the WD only increase based on the monthly linear increase in total precipitation.

### 3.3.3 Light, moderate and heavy rains

The impact of climate change on these indices is computed similar to the WD indices. As expected the projected changes obtained with the DC method are directly related to the change in mean seasonal rainfall with decreases in LRD (Fig. 8) and increases in both MRD and HRD (Figs. 9 and 10). Overall rainy days will become wetter with an exception the Delta region in MIROC-ESM that partially includes the sea. The increase in MRD and HRD is particularly visible during the long term climate projection (2080s). The general increase in HRD is confirmed by the results of the LS method applied to HadGEM2-ES and MIROC-ESM for the Mountainous region. Notable are the minimal

**HESSD**

doi:10.5194/hess-2015-499

**Downscaling GCM data for climate change impact assessments on rainfall**

R. J. Dahm et al.

Discussion Paper | Discussion Paper | Discussion Paper | Discussion Paper |

changes in HRD obtained from GFDL-CM3. This change in day-to-day variability is a direct result of changes in rain intensity and frequency in the GCM itself.

### 3.3.4 Return periods

Both HadGEM2-ES and MIROC-ESM project decreases in return periods for extreme rainfall amounts for both the Delta and Mountainous regions (see Fig. 11). In general changes derived after applying the DC method are more modest as a result of the unchanged day-to-day variability – changes will most likely be within the present day climate variability. Unfortunately, we find unrealistic high rainfall extremes for MIROC-ESM due to the high February–May multipliers that correct for the underestimation of winter and pre-monsoon rainfall in the Delta region. The possible increases in return periods for the delta region, obtained after applying the LS method to GFDL-CM3, is inconsistent with all other projections and is likely a result of correction multipliers below one that also decrease the extremes. The overall reduction in return times of annual extremes of daily rainfall for the late 21st century, which we find here, was also found by Ramesh and Goswami (2014).

The return period analysis once more illustrates the importance of focussing on relative changes. The precipitation amounts for a 100 year return period of a one-day rainfall event for the period 1990–2013 are 378, 328, and 229 mm for Akhuapada, Anandpur and Tilga rainfall gauge respectively, whereas the climatological data and the baseline rainfall for the LS method applied to all GCMs and regions show intensities of 100–200 mm for the same return period. Therefore, translating absolute future climate results of extreme rainfall events based on the climatological dataset to rainfall intensities recognizable for local authorities familiar with the rainfall gauges remains a challenge.

**HESSD**

doi:10.5194/hess-2015-499

**Downscaling GCM data for climate change impact assessments on rainfall**

R. J. Dahm et al.

## 4  Discussion

For this assessment the climatological dataset was the best available historical reference dataset as it provides continuous daily time-series for the full baseline period of the GCM simulations and performs relatively well for India (Rana et al., 2014). Yet our comparison with gauge observations showed large biases which are most likely as a result of spatial smoothening introduced by the interpolation to the grid. The biases in extreme rainfall amounts in the climatological dataset will have influenced the analysis of changes therein. In addition, with a more reliable reference meteorological dataset that better matches the CDF of gauge data, an advanced correction method could have been employed. With for example quantile-mapping realistic changes in extremes can be introduced and the day-to-day variability can be corrected towards observations (Van Pelt et al., 2012; Trenberth, 2011).

Instead we applied two relatively simple GCM correction methods. From both methods we know they have disadvantages for use in climate impact analysis. The DC method only provided reliable information on changes in seasonal means, as it leaves the day-to-day variability unchanged and changes in extremes are linearly scaled with changes in the mean (Camici et al., 2014; Chen et al., 2013; Teutschbein et al., 2012a). With the LS method we could obtain information on changes in rainfall frequency and intensity. Yet, as only monthly mean correction multipliers were applied, the day-to-day variability of corrected GCM baseline simulations remained different from the observed time-series. In addition, the monthly mean correction also influenced the values of the extremes, i.e. within an on average too wet month the extremes are linearly reduced with the same correction multiplier. Consequently, only relative changes in extremes could reliably be estimated from the baseline and future corrected GCM simulations.

We introduced an additional correction to the LS method by applying the multiplier for July to the data of June; this is to avoid the occurrence of extreme high rainfall amounts in the corrected data for June by compensating for the variable monsoon onset date. This introduces a deviation in the corrected baseline GCM data from observations and

Discussion Paper | Discussion Paper | Discussion Paper | Discussion Paper |

**HESSD**

doi:10.5194/hess-2015-499

**Downscaling GCM data for climate change impact assessments on rainfall**

R. J. Dahm et al.

rainfall amounts for June in both the corrected baseline and also the future climate GCM data and will remain too low. In future work the selection of GCMs could be extended with more criteria, focussing not only on annual rainfall amounts, but also on the onset and amount of monsoon rainfall. Hereby the need to apply such tailored corrections can be avoided.

From the above we can conclude that the performed analysis is not based upon the latest state-of-the-art techniques as for example presented in Teutschbein and Seibert (2012b), Chen et al. (2012), Christensen et al. (2008), Fowler et al. (2007), Van Pelt et al. (2012) and many others. Yet this study illustrates the restrictions one is faced with when working in less data rich areas. As discussed, the quality in combination with the time-coverage of the historical observations did neither allow for statistical down-scaling based upon predictor–predictant relationships nor for more advanced quantile mapping techniques. In addition, dynamically downscaled RCM data was not at hand. The simpler techniques that were applied did not allow for an analysis of all statistical quantities of interest. This demonstrates the gap between science and practice that still needs to be bridged in the area of local climate impact analysis (Ehret et al., 2012; Hagemann et al., 2011; Dosio et al., 2012).

This study focused on changes in rainfall only but could be extended with impact analysis for water resources and flood risk management. As discussed above the changes in daily rainfall and extreme events cannot be considered reliable, yet the seasonal and monthly changes provide information for (1) flood risk assessments for larger river basins where severe floods mainly occur as multiple day events due to re-sult of medium/long-term (weeks to months) rainfall conditions upstream and for (2) water resources management, groundwater recharge and soil moisture conditions of relevance to agricultural production. For the latter application the baseline change as-sessment should be extended to all seasons.

# 5  Conclusions

– SW monsoon rainfall is in general projected to increase over the Brahmani-Baitarani river basin, especially in the Mountainous regions.

– The number of wet days is likely to increase in the Monsoon season; changes are largest and most pronounced towards the end of the century. The number of days with heavy rain is also likely to increase.

– Although the annual rainfall cycle is captured well by all three GCMs, biases from observed data exist, with as most important the too late onset of the monsoon in HADGEM2-ES and MIROC-ESM and the underestimation of winter rainfall by MIROC-ESM in the delta region.

– The DC and LS correction methods disturb the projected changes in extremes. The LS method is preferred when analysing extremes and change in number of wet days, as the DC method leaves the day-to-day variability unchanged. Yet with the LS method the analysis should be restricted to the relative changes between corrected baseline and future GCM time-series.

– The climatological observation dataset APHRODITE underestimates the number of "heavy rain" events when compared to observed data of CWC rain gauge stations. This affects the reliability of future climate change projections for precipitation.

– For proper correction and downscaling of GCM data more advanced techniques exist. However, these techniques require high quality reference meteorological datasets and/or computing resources which are sometimes not available for practical analysis in the developing countries. Assessment of the impact of these limitations to the actual climate change analysis will provide insight in the applicable bias correction methods and can improve the interpretation of the results.

Discussion Paper | Discussion Paper | Discussion Paper | Discussion Paper | Discussion Paper |

**HESSD**

doi:10.5194/hess-2015-499

**Downscaling GCM data for climate change impact assessments on rainfall**

R. J. Dahm et al.

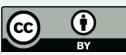

*Acknowledgements.* The study was funded by ADB and DFID/UKAid under the Asian Development Bank Policy and Advisory Technical Assistance 8089 IND Phase II project. We acknowledge our colleagues of the TA8089 project and we would like to express our sincere thanks to S. Sethurathinam as well as Central Water Commission officers Vasanthakumar Venkatesan and Manoj Kumar. We acknowledge the World Climate Research Programme's Working Group on Coupled Modelling, which is responsible for CMIP, and we thank the climate modelling groups (listed in Table 1 of this paper) for producing and making available their model output. For CMIP the US Department of Energy's Program for Climate Model Diagnosis and Intercomparison provides coordinating support and led development of software infrastructure in partnership with the Global Organization for Earth System Science Portals.

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

**Table 1.** List of GCMs used in this study.

| Institute | GCM | Spatial resolution | Emission scenario |
|---|---|---|---|
| Hadley Center (UK) | HadGEM2-ES | $1.87° \times 1.25°$ | RCP6.0 |
| Japan Agency for Marine-Earth Science and Technology, Atmosphere and Ocean Research Institute (The University of Tokyo), and National Institute for Environmental Studies | MIROC-ESM | $2.81° \times 2.79°$ | RCP6.0 |
| National Oceanic and Atmospheric Administration (United States) | GFDL-CM3 | $2.5° \times 2.0°$ | RCP6.0 |

**HESSD**

doi:10.5194/hess-2015-499

**Downscaling GCM data for climate change impact assessments on rainfall**

R. J. Dahm et al.

**Table 2.** Latitude and longitude of the CWC rain gauge stations and nearest climatological dataset (APHRODITE) center grid cell point.

| Name | Geographic location | Rain gauge station | | APHRODITE | |
|---|---|---|---|---|---|
| | | Latitude (°N) | Longitude (°E) | Latitude (°N) | Longitude (°E) |
| Tilga | Mountainous | 22°37' | 84°24' | 22°22' | 84°22' |
| Anandpur | Delta | 21°12' | 86°7' | 21°7' | 86°22' |
| Akhuapada | Delta | 20°54' | 86°16' | 20°52' | 86°22' |

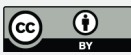

**HESSD**

doi:10.5194/hess-2015-499

**Downscaling GCM data for climate change impact assessments on rainfall**

R. J. Dahm et al.

**Table 3.** Indian Meteorological Department classification of rainfall intensity.

| Event | Abbreviation | Rainfall |
|---|---|---|
| Wet Day | WD | Daily rainfall $\geq$ 2.5 mm |
| Light rain | LRD | Daily rainfall between 2.5 and 7.5 mm |
| Moderate rain | MRD | Daily rainfall between 7.6 and 35.5 mm |
| Rather heavy rain | HRD | Daily rainfall between 35.6 and 64.4 mm |
| Heavy rain | | Daily rainfall between 64.5 and 124.4 mm |
| Very heavy rain | | Daily rainfall between 124.5 and 244.4 mm |
| Extremely heavy rain | | Daily rainfall > 244.5 mm |

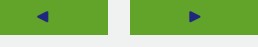

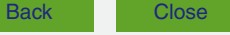
Interactive Discussion

**HESSD**

doi:10.5194/hess-2015-499

**Downscaling GCM data for climate change impact assessments on rainfall**

R. J. Dahm et al.

**Table 4.** Classification of seasons.

| Season | Months | Abbreviation |
|---|---|---|
| Winter | Dec, Jan, Feb | DJF |
| Pre-monsoon | Mar, Apr, May | MAM |
| SW monsoon | Jun, Jul, Aug, Sep | JJAS |
| Post monsoon | Oct, Nov | ON |

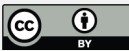

**HESSD**

doi:10.5194/hess-2015-499

**Downscaling GCM data for climate change impact assessments on rainfall**

R. J. Dahm et al.

**Table 5.** SRTM30 elevation of center grid points (in m + MSL).

| Region | Data source | HadGEM2-ES | MIROC-ESM | GFDL-CM3 |
|--------|-------------|------------|-----------|----------|
| Delta | GCM | 257 | 0 | 29 |
| | Observation | 27 | 2 | 13 |
| Mountainous | GCM | 375 | 665 | 888 |
| | Observation | 353 | 503 | 506 |

Discussion Paper | Discussion Paper | Discussion Paper | Discussion Paper

## HESSD

doi:10.5194/hess-2015-499

**Downscaling GCM data for climate change impact assessments on rainfall**

R. J. Dahm et al.

**Table 6.** Original average correction factors for the month June. Italic values are $\geq 1.5\times$ the July correction factor, and bold values are $\geq 3\times$ the July correction factor.

| Region | Method | Reference | HadGEM2-ES | MIROC-ESM | GFDL-CM3 |
|---|---|---|---|---|---|
| Delta | LS | – | **8.28** | **7.34** | 0.75 |
| | DC | 2040 | *1.87* | **8.54** | 0.99 |
| | | 2080 | 1.09 | **4.57** | 0.97 |
| Mountainous | LS | – | **11.1** | *3.85* | 1.30 |
| | DC | 2040 | *1.95* | *2.69* | 0.98 |
| | | 2080 | *2.33* | 1.69 | 1.17 |

# HESSD

doi:10.5194/hess-2015-499

**Downscaling GCM data for climate change impact assessments on rainfall**

R. J. Dahm et al.

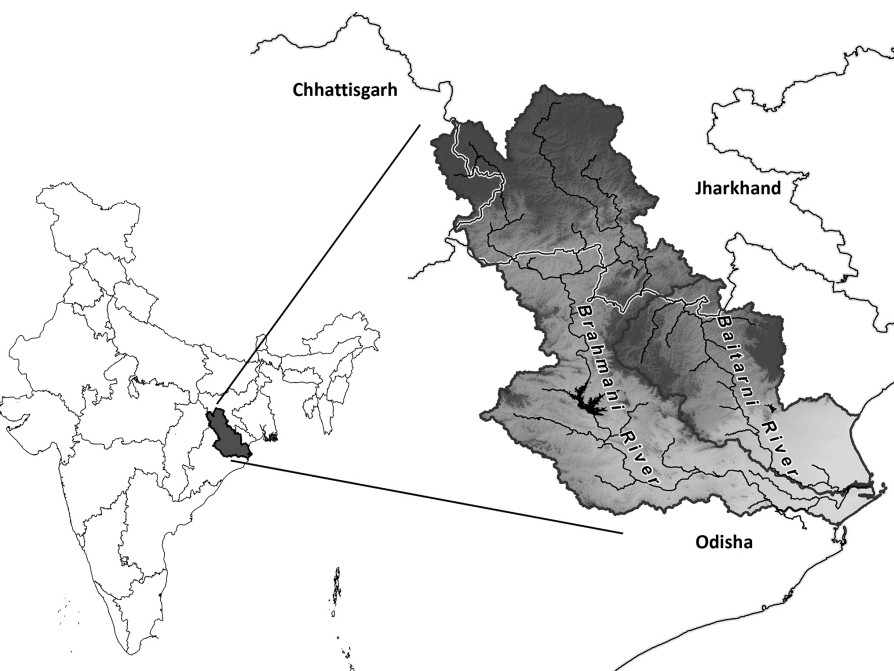

**Figure 1.** Map showing location of the Brahmani-Baitarani river basin as study area and the boundaries of the three surrounding Indian states.

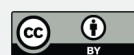

**HESSD**

doi:10.5194/hess-2015-499

**Downscaling GCM data for climate change impact assessments on rainfall**

R. J. Dahm et al.

**Figure 2.** Locations of the GCMs center points (in black): HadGEM2-ES (square), MIROC-ESM (triangle), and GFDL-CM3 (round). The nearest climatological dataset center points have the same marker (in grey). CWC rain gauges are shown with station name (black star).

Discussion Paper | Discussion Paper | Discussion Paper | Discussion Paper | Discussion Paper |

# HESSD

doi:10.5194/hess-2015-499

**Downscaling GCM data for climate change impact assessments on rainfall**

R. J. Dahm et al.

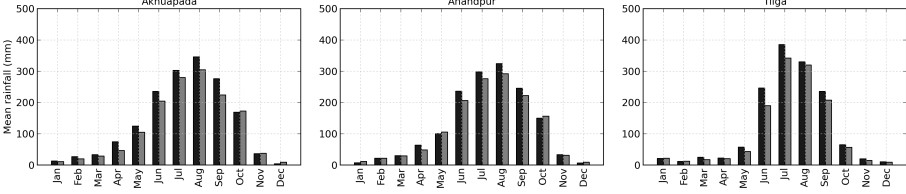

**Figure 3.** Monthly mean precipitation for the period 1990–2007 as observed in the climatological dataset (grey) and by CWC (black).

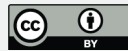

Discussion Paper | Discussion Paper | Discussion Paper | Discussion Paper | Discussion Paper |

**HESSD**

doi:10.5194/hess-2015-499

**Downscaling GCM data for climate change impact assessments on rainfall**

R. J. Dahm et al.

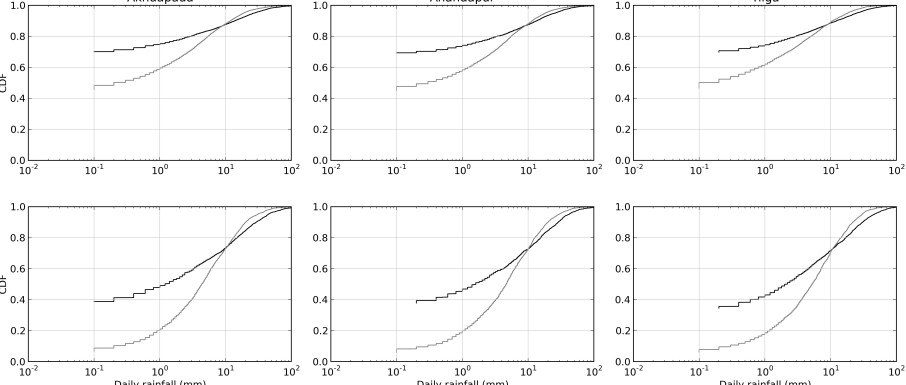

**Figure 4.** CDF curves of daily rainfall for the period 1990–2007 as observed in the climatological dataset (grey) and by CWC (black) for the entire year (top) and monsoon season (bottom).

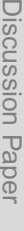

# HESSD

doi:10.5194/hess-2015-499

**Downscaling GCM data for climate change impact assessments on rainfall**

R. J. Dahm et al.

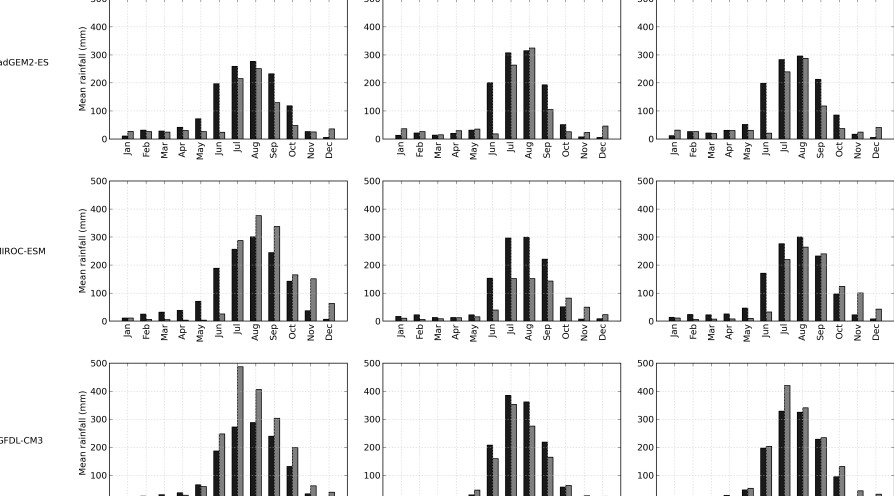

**Figure 5.** Monthly mean rainfall for the period 1961–1990 as observed (black) and simulated by the three GCMs (grey).

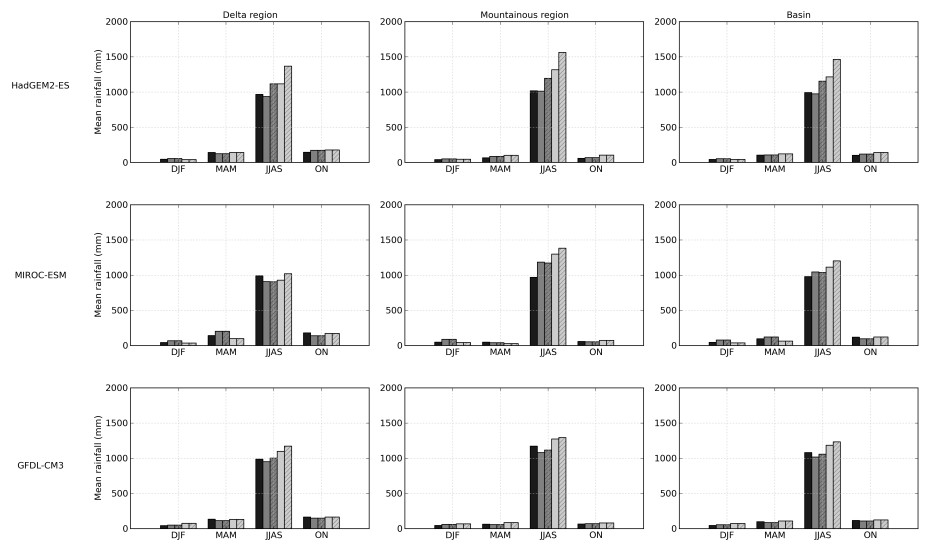

**Figure 6.** Mean seasonal rainfall. Observations (black), 2040 short term climate projection (dark grey: LS, dashed dark grey: DC), and 2080 long term climate projection (light grey: LS, dashed light grey: DC).

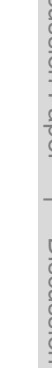

# HESSD

doi:10.5194/hess-2015-499

**Downscaling GCM data for climate change impact assessments on rainfall**

R. J. Dahm et al.

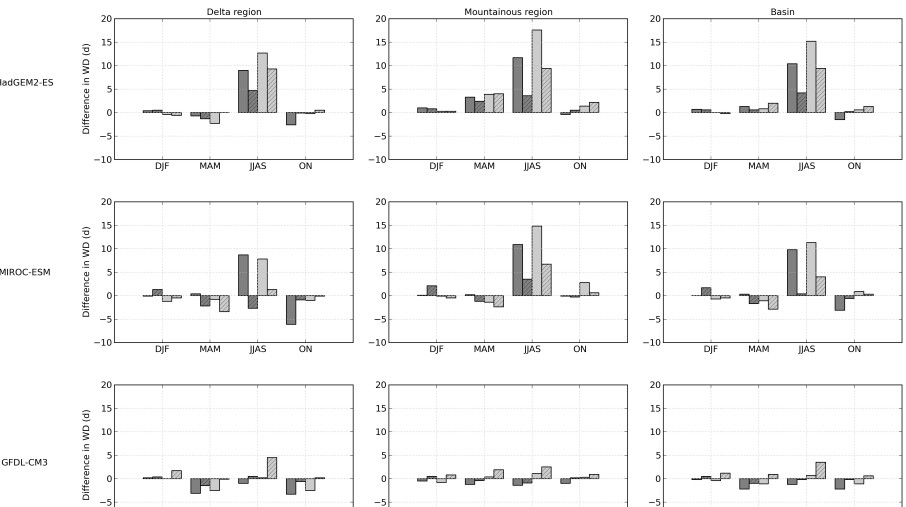

**Figure 7.** Difference in number of Wet Days: 2040 short term climate projection (dark grey: LS, dashed dark grey: DC), and 2080 long term climate projection (light grey: LS, dashed light grey: DC).

Discussion Paper | Discussion Paper | Discussion Paper | Discussion Paper | Discussion Paper

**HESSD**

doi:10.5194/hess-2015-499

**Downscaling GCM data for climate change impact assessments on rainfall**

R. J. Dahm et al.

# HESSD

doi:10.5194/hess-2015-499

**Downscaling GCM data for climate change impact assessments on rainfall**

R. J. Dahm et al.

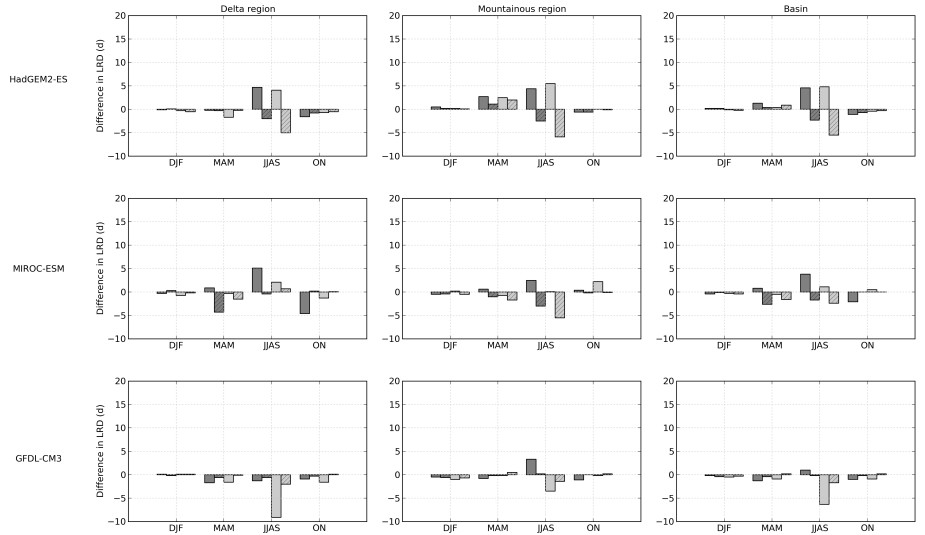

**Figure 8.** Difference in number of LRD: 2040 short term climate projection (dark grey: LS, dashed dark grey: DC), and 2080 long term climate projection (light grey: LS, dashed light grey: DC).

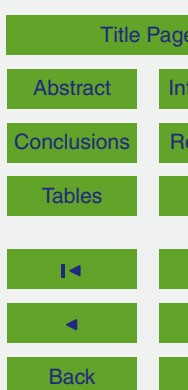

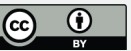

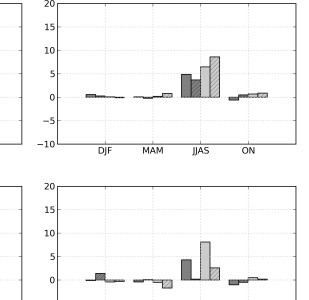

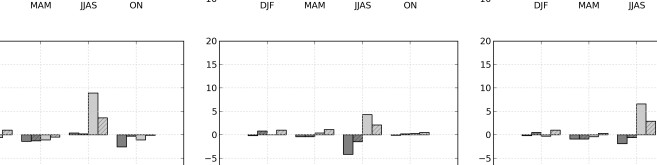

**Figure 9.** Difference in number of MRD: 2040 short term climate projection (dark grey: LS, dashed dark grey: DC), and 2080 long term climate projection (light grey: LS, dashed light grey: DC).

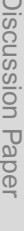

# HESSD

doi:10.5194/hess-2015-499

**Downscaling GCM data for climate change impact assessments on rainfall**

R. J. Dahm et al.

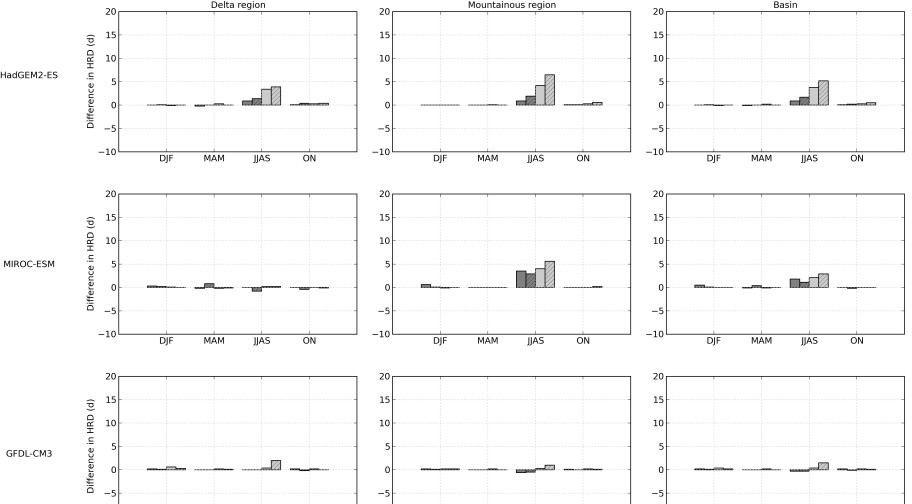

**Figure 10.** Difference in number of HRD: 2040 short term climate projection (dark grey: LS, dashed dark grey: DC), and 2080 long term climate projection (light grey: LS, dashed light grey: DC).

Discussion Paper | Discussion Paper | Discussion Paper | Discussion Paper | Discussion Paper |

**HESSD**

doi:10.5194/hess-2015-499

**Downscaling GCM data for climate change impact assessments on rainfall**

R. J. Dahm et al.

# HESSD

doi:10.5194/hess-2015-499

**Downscaling GCM data for climate change impact assessments on rainfall**

R. J. Dahm et al.

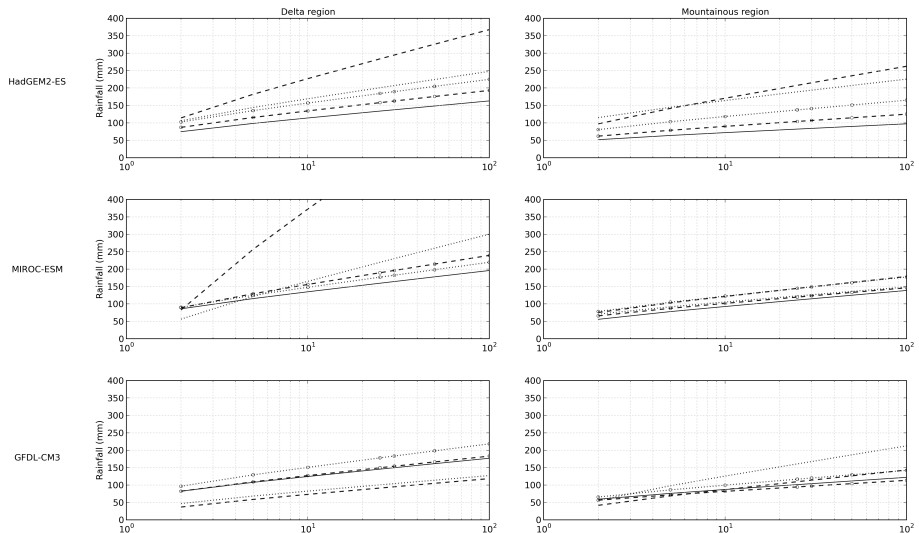

**Figure 11.** Return periods based on annual maximum. Observations (black), 2040 short term climate projection (dashed: LS, dashed with circle: DC), and 2080 long term climate projection (dotted: LS, dotted with circle: DC).