# Peer review of "Downscaling GCM data for climate change impact assessments on rainfall: a practical application for the Brahmani-Baitarani river basin"

_Hydrology and Earth System Sciences, 2015_

## Referee Comment (RC1) · Anonymous Referee #1 · 23 Feb 2016

**General comments**

This study explores changes in precipitation over the Brahmani-Baitarani river basin (51 822 km$^2$) in India using three GCMs runs from CMIP5 and two basic bias-correction methods (delta change and linear scaling). Two key challenges that the study is addressing are the low station density and the lack of dynamically downscaled climate simulations. This paper provides interesting insights into challenges faced in regions of low data availability in the context of impact modeling. It could formulate general recommendations on the selection of bias-correction methods in those regions, but to

achieve this goal, I consider that further research and major modifications are necessary.

The authors rule out bias-correction methods more complex than delta change and linear scaling because of low data availability. The fundamental question this choice poses is: "What are the data requirements for different bias-correction methods, and in particular, how long should time series be?". I argue that this question should be carefully addressed by the authors on the basis of i) the existing literature (e.g., Rajczak et al. (2015) and references therein), ii) general statistical considerations (e.g., how many data points are required to constrain the distribution of the tails, which can then be used for bias-correction) and possibly, iii) by conducting further numerical experiments (e.g. illustrate the risks of applying complex bias-correction methods with insufficient data by highlighting suspicious/unrealistic features in postprocessed time series).

Overall, the way the authors deal with extreme events lacks consistency. On one hand, they exclude a large fraction of the available bias-correction methods because of low data availability (P8L6-10). On the other hand, they use the same observations to fit a Gumbel distribution and estimate rainfall intensities associated with return periods of up to 100 years (P10L10, I suggest by the way that the authors provide uncertainty estimates of the intensity of these rare events). If this can be done, why not use this information to bias-correct GCM simulations? Similarly, the authors acknowledge that the postprocessing methods they selected "only concentrate on the correction of monthly mean rainfall amounts and can thus affect our analysis of changes in extreme rainfall indices". I certainly understand that they are limited by the data, but I urge them to critically assess and discuss how much can be learned about extreme events when methods such as delta change and linear scaling are used. It is really a matter of communicating meaningful information, especially when the results are then used for decision making.

One way to overcome the lack of data is to make stronger links to processes leading to floods (see the related discussion on the consideration of misrepresented process

when bias-correcting models in Addor et al., 2016). It means identifying those processes (e.g. storm surges described on P10L16-23 or specific aspects of the monsoon), evaluate how they are captured by the models, and then, instead of simply looking at how precipitation is projected to change, look at how these processes are projected to change. Their scale is probably large enough to be captured by GCMs. If this link is made, then the low station density in the region would be less problematic and the study could be a major breakthrough.

**Specific comments**

The authors "focus on three models that nearly span the full range of annual mean rainfall projections available for India" (P4L19-20). Please also comment on how well they cover the range of projected changes in extreme events, which are also a main focus of the study.

P5L18-19: Why are the authors using 30 years for the reference period and then 40 years for the future conditions? This should be harmonized.

Section 2.3: Please briefly explain how orographic effects were accounted for when generating the APHRODITE data set. Were measurements from the three CWC rain gauges also used to produce the APHRODITE data set? Which period is covered by the three gauges? The authors write that "gauge observations could not be used as reference since they do not overlap with the full GCM baseline period" (P7L6-7), please develop.

P12L5-15: The differences reported between the two observational datasets are indeed large, it is in particular clear from Figure 4, although it is not completely surprising since the CWC curves are produced using solely three stations for 51 822 km$^2$. Yet I think the observational uncertainty should be better accounted for and displayed, as it influences the bias-correction and thereby the projected changes. Also, I understand that the "climatological data set" is APHRODITE, if it is the case, I would state it more clearly.

**Technical corrections**

P5L20-21: "Almost all CMIP5 models show good performance for surface air temperature simulations averaged over South Asia and the Indian Sub-continent", please add a reference.

P4L9: need

P6L11-12: I suggest using "wide" instead of "full"

P6L19: What is Monsoon Asia?

P7L20: Out of curiosity: were RCMs ever run over India, for instance under SRES emission scenarios, and if yes, please provide references and discuss what this revealed.

P13L14: 900mm/month?

General comments on the figures: I encourage the authors to use colors, that is free of charge when publishing in HESS, and it would make their figures easier and more enjoyable to read. I recommend using full names in the captions instead of acronyms (e.g. MRD, HRD) and using a larger font size.

Figure 11: this figure definitively needs uncertainty bounds.

**Suggested references**

Addor, N., Rohrer, M., Furrer, R. and Seibert, J.: Propagation of biases in climate models from the synoptic to the regional scale: Implications for bias adjustment, J. Geophys. Res. Atmos., doi:10.1002/2015JD024040, 2016.

Rajczak, J., Kotlarski, S., Salzmann, N. and Schär, C.: Robust climate scenarios for sites with sparse observations: a two-step bias correction approach, Int. J. Climatol., doi:10.1002/joc.4417, 2015.

---

## Referee Comment (RC2) · Anonymous Referee #2 · 25 Feb 2016

Downscaling GCM data for climate change impact assessments on rainfall: a practical application for the Brahmani-Baitarani river basin

By R. J. Dahm, U. K. Singh, M. Lal, M. Marchand, F. C. Sperna Weiland, S. K. Singh, and M. P. Singh

In this paper, the authors perform a climate change analysis for the Brahmani-Baitarani river basin in India. For this purpose, the authors use three GCMs. Further, because of absence of reliable long historic meteorological data in the study area, they use two easy to implement bias correction approaches. The authors conclude that in general

the south west monsoon rainfall, number of wet days and number of days with heavy rain are projected to increase over the study area. However, the authors also state that the APHRODITE dataset which is considered as the observed meteorological dataset underestimates the number of "heavy rain" events when compared to observed data of Central Water Commission (CWC) rain gauge stations and that this will affect the reliability of future climate change projections for precipitation. The topic is very relevant and such a study is especially useful for developing countries like India where data availability is always scarce. However, the current version of the manuscript neither makes any contribution in terms of the use of advanced state-of-art techniques nor does it provide any new insight for the study area. Therefore, in my opinion, the manuscript in its present state does not make a significant contribution to justify its publication. I therefore would suggest a couple of major revisions to improve the manuscript.

(i) The authors use APHRODITE data as the observed climatological rainfall dataset stating that it includes observations from over 2000 stations over India and captures the large scale features of monsoon rainfall over the Indian region well (Rajeevan and Bhate, 2008). In addition they also use observed rainfall dataset from three raingauge stations of CWC. However, for the study area, the authors could well have used a daily gridded rainfall dataset which is developed by the India Meteorological Department (IMD) (Pai et al., 2014). This is a high spatial resolution ( $0.25 \times 0.25$  degree) dataset available for a long period (1901–2010) and has been developed using daily rainfall records from 6955 rain gauge stations all over India. Quite a few researchers have established that this IMD product is much more accurate than the APHRODITE dataset. In addition, station rainfall data for the study area is also available from IMD. There are quite a few raingauge stations in the study area (in addition to the three raingauge stations of CWC) for which daily rainfall data for 25 to 30 years are available from IMD. The authors may consider using these datasets. Thus, the study area is not as data scarce as has been made out to be by the authors.

However, given this situation, authors have a very good scope of improving their manuscript by considering different data availability scenarios. For example the authors may consider (i) availability of only IMD station raingauge dataset, (ii) only IMD gridded rainfall dataset (iii) only APHRODITE dataset (as they have done in the present manuscript) etc. This way the authors can test which advanced bias correction technique works best under different data availability scenarios. This would be a good contribution for data scarce developing countries.

Pai, D.S., Sridhar, L., Rajeevan, M., Sreejith, O.P., Satbhai, N.S., Mukhopadhyay, B., 2014. Development of a new high spatial resolution ( $0.25 \times 0.25$ ) long period (1901–2010) daily gridded rainfall data set over India and its comparison with existing data sets over the region. Mausam 65, 1–18.

(ii) In general, the authors presented an increase/decrease or underestimation/overestimation in a given statistic. The authors should have also done checks on whether the increase/decrease or underestimation/overestimation is statistically significant. Also, the conclusions of the study (i.e. projected increase of SW monsoon rainfall, number of wet days and number of days with heavy rain over the study area) are more or less in line with what has been stated by previous researchers for the study area or the eastern region of India. So I do not see any new contribution here. But the reasons for such changes should be investigated and this could be a meaningful contribution.

Some of the important citations like Chaturvedi et al., 2012; Thrasher et al., 2012; Rana et al., 2014 are missing in the reference list.

---

## Author Comment (AC1)

**Reply to review comments Referee #2**

*We would like to thank the reviewer for her/his extensive review of our manuscript in the HESS-category 'Cutting-edge case studies' and the sound topics brought forward. To a certain extent we agree with the statement that the study area is not as data scarce as presented. In this reply we explain why actual data scarcity still was a serious issue for us in this study. It even set limitations for the bias correction approaches that could be selected. Here we also want to emphasize that as hydrologists we are forced to use the data that have been provided in a project, that are available for a reasonable price or ideally datasets that are open source. See also our comments to the remark where the use of the Indian Meteorological Department (IMD) dataset is mentioned for the first time (page 2). In the following sections we reply (in italics) to all of the reviewers comments (cited):*

**General comments**
"In this paper, the authors perform a climate change analysis for the Brahmani-Baitarani river basin in India. For this purpose, the authors use three GCMs. Further, because of absence of reliable long historic meteorological data in the study area, they use two easy to implement bias correction approaches. The authors conclude that in general the south west monsoon rainfall, number of wet days and number of days with heavy rain are projected to increase over the study area. However, the authors also state that the APHRODITE dataset which is considered as the observed meteorological dataset underestimates the number of "heavy rain" events when compared to observed data of Central Water Commission (CWC) rain gauge stations and that this will affect the reliability of future climate change projections for precipitation."

> *Indeed our analysis indicated that the heavy rain events are underestimated by the APHRODITE dataset. For the estimation of future changes the Delta Change (DC) and Linear Scaling (LS) methods have been applied to the same slightly biased dataset – we agree that it is therefore likely that the absolute quantification of heavy rain events will also be biased in the future projections. Yet, the relative changes are expected to be of the same order of magnitude.*

> *To check this we applied the DC-multipliers of the three GCMs for 2045s and 2085s to the CWC station observations and compared the relative changes. The statistics of mean seasonal rainfall, number of wet days (WD), and the number of days with light and moderate rain (LRD, MRD) in general show a similar direction and order of magnitude. The change statistics of the number of days with heavy rain (HRD) show no clear direction.*

"The topic is very relevant and such a study is especially useful for developing countries like India where data availability is always scarce. However, the current version of the manuscript neither makes any contribution in terms of the use of advanced state-of-art techniques nor does it provide any new insight for the study area. Therefore, in my opinion, the manuscript in its present state does not make a significant contribution to justify its publication. I therefore would suggest a couple of major revisions to improve the manuscript."

> *We agree with the reviewer that we did not apply advanced state-of-the-art techniques for bias correction. The two-fold reason for that approach is discussed in the above reply. To the knowledge of the authors no dedicated study of the impacts of climate change on rainfall in the Brahmani-Baitarani River basin has been published in peer-reviewed international journals. The manuscript therefore provides additional insight for that specific region in India. This region is of special interest because it encompasses both coastal and mountainous areas that in this specific case show a mixed response to climate change. We will add the following sentence on the specific spatial focus of this study to the introduction:*

>> "This basin is interesting because it encompasses both coastal and mountainous areas that will likely show a mixed response to climate change" (P4L5).

> *Additionally, as a cutting-edge case study the manuscript shows what choices one has to deal with when carrying out this type of climate change assessment studies, e.g. when some data is just not available to the researchers.*

*Most peer-reviewed papers on advanced down-scaling and bias-correction methods address the ideal situation where long historic observational records are available, yet in reality these data may not be present. As reviewer 1 suggested we have read the recent paper of Rajczak et al. (2015). We do appreciate the idea of spatially transferring bias-correction functions. However, Rajczak et al. already mention the limitations for spatial heterogeneous variables such as precipitation especially when the reference datasets are short. For the derivation of representative transfer functions the calibration is ideally based on long-term data records, this to include the natural variability sufficiently with anomalous wet and dry periods. Rajczak et al. (2015) suggest the use of a climatological representative period of at least 20 years. Within their study they show that by using only a limited number of years, for the calibration of a Quantile Mapping transfer function, large biases in absolute monthly precipitation amounts can be obtained.*

*This confirms the statements we made that an advanced method such as quantile mapping cannot be applied in our situation since we do not have rain gauge observed data and GCM data that overlap in time and the observational records only have a limited length. Differences in duration curves may as well occur due to trends in precipitation or incomplete inclusion of natural variability. We add the following comment to the manuscript (P8L3):*

> "For the derivation of representative transfer functions for the Quantile Mapping approach the calibration is usually based on long-term data records, ideally more than 20 years, this to include the natural variability sufficiently with anomalous wet and dry periods (Rajczak et al., 2015). Using only a limited number of (overlapping) years, which would be the case if we use the CWC gauge data as reference, can result in large biases in absolute monthly rainfall amounts. When working with non-overlapping periods, an additional difference can be introduced by trends in rainfall time-series and according to Goswami et al. (2012) there has been an increasing trend in both frequency and intensity of heavy rain events over Central India of 10% per decade since 1950. Moreover the Quantile Mapping method requires reliable CDFs of reference rainfall and from the comparison between the APHRODITE dataset and station observations (Rana et al., 2014) we know that large biases in rainfall extremes exist indicating that the APHRODITE dataset cannot be used as reference for Quantile Mapping."

"The authors use APHRODITE data as the observed climatological rainfall dataset stating that it includes observations from over 2000 stations over India and captures the large scale features of monsoon rainfall over the Indian region well (Rajeevan and Bhate, 2008). In addition they also use observed rainfall dataset from three raingauge stations of CWC. However, for the study area, the authors could well have used a daily gridded rainfall dataset which is developed by the India Meteorological Department (IMD) (Pai et al., 2014). This is a high spatial resolution (0.25× 0.25 degree) dataset available for a long period (1901–2010) and has been developed using daily rainfall records from 6955 rain gauge stations all over India. Quite a few researchers have established that this IMD product is much more accurate than the APHRODITE dataset."

> *In the manuscript acknowledgements it is stated that we received funding from both the Asian Development Bank (ADB) and DFID/UKAid. The case study was carried out in cooperation with the Indian Central Water Commission (CWC) under the Technical Assistant program of the ADB. The reviewer refers to an IMD gridded data set which is indeed available. The development of that data set is discussed in Pai et al. (2014). Pai et al. (2014) describes the high spatial resolution (0.25° × 0.25°), daily gridded rainfall data set (1901-2010) as developed by IMD.*
> *We considered using this dataset for the study. The data is not yet available free of any costs and the conditions of the Indian Meteorological Department to make this data available made it impossible for us to purchase it. IMD requests to sign for the following: "The data are meant exclusively for your own use and shall not be passed on to any other party or agency (Indian or Foreign) either in part or in full. If so needed, prior approval in writing will be taken from the*

India Meteorological Department for the same. The data shall not be used for any commercial purpose or to earn consultancy fees, honoraria etc." *Because this case study was carried out as part of a consultancy-based TA-project for the ADB it was impossible for us to purchase the IMD gridded data. Our local Indian consortium partner tried as well but did not succeed. This indeed had implications for the analysis, but was beyond the researchers' control. This is the very reason why we mentioned in our manuscript that 'data and model availability can be limited […] because political data sensitivity hampers open sharing' (P2L9-11).*

*This leaves us to check whether there is information available on the performance of APHRODITE in comparison to the IMD data set. Pai et al. (2014) compared the gridded IMD data set (i.e. IMD4) with other previous IMD gridded data sets and the APHRODITE data set. The comparison suggests that "the climatological and variability features of rainfall over India derived from IMD4 were comparable with the existing gridded daily rainfall data sets." The all mean annual rainfall and SW Monsoon rainfall in IMD4 was found approx. 13% and 11% respectively wetter than in APHRODITE, with a 0.95 correlation coefficient for both assessments. We include the following:*

> "Additionally, and similar to the findings when comparing CWC gauge rainfall to APHRODITE, Pai et al. (2014) concluded IMD4 to be approximately 13% and 11% wetter in mean annual rainfall and SW monsoon rainfall than the APHRODITE gridded rainfall." (P12: Section 3.1)

*Prakash et al. (2014) have shown that APHRODITE and GPCC performed best among four gridded rain gauge-based land-only rainfall products which all are open available (CPC, CRU, GPCC, and APHRODITE) when compared with IMD generated gridded rainfall data. Hence, the gridded APHRODITE rainfall data set was the best open available data set to us.*

*We add a comment to the manuscript on the availability of the IMD gridded data set and the findings of Prakash et al. (2014) to the introduction.*

> "The Indian Meteorological Department has developed a daily gridded rainfall data set (1901-2010) in high spatial resolution of $0.25° \times 0.25°$ (Pai et al., 2014). This would have been the ideal dataset for the analyses in this study as it contains data from 6955 gauges over India and covers the period 1901-2100. Yet, due to restrictive conditions set by IMD this data set could not be purchased and applied in this study. We therefore used the APHRODITE dataset. Pai et al. (2014) showed the comparability of the APHRODITE data set regarding the climatological and rainfall variability over all India. Prakash et al. (2014) listed APHRODITE together with Global Precipitation Climatology Center (GPCC) product as the best performing among 4 rain gauge-based land-only rainfall data sets when compared to the IMD gridded data set." (P4L13)

"In addition, station rainfall data for the study area is also available from IMD. There are quite a few raingauge stations in the study area (in addition to the three raingauge stations of CWC) for which daily rainfall data for 25 to 30 years are available from IMD. The authors may consider using these datasets. Thus, the study area is not as data scarce as has been made out to be by the authors. "

> *We agree with the reviewer that these raingauge stations exist. During the study we asked IMD to provide this dataset. IMD provided us a quote for an average of 100 USD/year/raingauge. With e.g 10 raingauges and 30 year of data that would have cost ~30KUS$. Unfortunately, limited available financial resources did not allow purchasing the datasets that we as hydrologists would have wanted to use. We acknowledge the fact that we should mention the data availability - meaning not the actual data presence but rather the possibility to use the data within the project - better in our manuscript. Unfortunately, this is reality in many projects*

*We add a comment to the manuscript on the availability of the IMD raingauge data in the method section.*

> "Several IMD rainfall gauges are present in the study area. However this data was not made
>
> available to this study."

"However, given this situation, authors have a very good scope of improving their manuscript by considering different data availability scenarios. For example the authors may consider (i) availability of only IMD station raingauge dataset, (ii) only IMD gridded rainfall dataset (iii) only APHRODITE dataset (as they have done in the present manuscript) etc. This way the authors can test which advanced bias correction technique works best under different data availability scenarios. This would be a good contribution for data scarce developing countries. "

> *We fully agree with the reviewer that when the IMD data would have been available to us this would have been a sound alternative scope for our manuscript. If the IMD gridded rainfall or raingauge data had been available to us, of course we would have used that data. Unfortunately, as this data was not available to us during the TA-project we cannot follow-up on the sound suggestion of the reviewer. The comment of the reviewer is exactly one of the main messages we wanted to address, how to deal with these kind of practical issues one will face during climate change assessments.*

"Pai, D.S., Sridhar, L., Rajeevan, M., Sreejith, O.P., Satbhai, N.S., Mukhopadhyay, B., 2014. Development of a new high spatial resolution (0.25× 0.25) long period (1901– 2010) daily gridded rainfall data set over India and its comparison with existing data sets over the region. Mausam 65, 1–18. (ii)"

> *We appreciate sharing this reference and we include it in the manuscript.*

"In general, the authors presented an increase/decrease or underestimation/overestimation in a given statistic. The authors should have also done checks on whether the increase/decrease or underestimation /overestimation is statistically significant."

> *We agree with the reviewer and have carried out t-Tests (one-tailed distribution, two-sample unequal variance for a significance level of $\rho = 0.05$) for each of the statistics. We add the following comments to the manuscript:*

> > "In general, all changes in mean monsoon rainfall derived with HadGEM2-ES and MIROC-ESM
> >
> > are significant ($\rho = 0.05$ level). GFDL-CM3 shows this significance only for the changes at the end
> >
> > of the century (2085s). For the mountainous region changes in pre-monsoon mean rainfall are also
> >
> > significant at $\rho = 0.05$ level for the long term climate projection." (Section 3.3.1 Mean seasonal
> >
> > rainfall)

> > "The changes in WD during monsoon due to climate change are significant ($\rho = 0.05$ level) for
> >
> > HadGEM2-ES and MIROC-ESM, except for the DC method in the Delta region in the latter GCM.
> >
> > GFDL-CM3 shows a more divers pattern with only significant changes of WD in the monsoon
> >
> > period for the long term climate projection (2085s)." (Section 3.3.2 Number of wet days)

> > "There is a strong indication for the impact of climate change during monsoon on LRD, MRD, and
> >
> > HRD in both regions at $\rho = 0.05$ significance level. (Section 3.3.3 Light, moderate and heavy rains)

"Also, the conclusions of the study (i.e. projected increase of SW monsoon rainfall, number of wet days and number of days with heavy rain over the study area) are more or less in line with what has been stated

by previous researchers for the study area or the eastern region of India. So I do not see any new contribution here. But the reasons for such changes should be investigated and this could be a meaningful contribution."

> *We appreciate the reviewer for noticing the potential of our Cutting-edge case study manuscript and mentioning it as a meaningful contribution. We would like to refer to the statement made earlier that the authors are not aware of studies on climate change impact on rainfall for the Brahmani-Baitarani River basin published in peer-reviewed international journals using the CMIP5 GCM experiments.*

> *We agree with the reviewer that many studies on climate change in India have been published, and some of them even focussing on the Brahmani River basin or the Baitarani River basin. Gosain et al. (2006) describe in Current Science a climate change impact analysis on hydrology of Indian river basins including the Brahmani River basin. A single transient climate experiment (HadRM2) has been applied. No detailed results on the Brahmani River basin are described. Islam et al. (2012) published in Water Resources Management the results of a study of streamflow response to climate change in the Brahmani River basin by a temperature and rainfall sensitivity analysis. No GCM experiments have been used. Mitra and Mishra (2014) published in the Journal of Indian Water Resources Society a hydrological impact assessment of the Baitarani River basin based on climate change sensitivity by changing the daily rainfall with ±5, 10, 15, and 20%. No GCM experiments have been used. Based on the above we conclude that for the Brahmani-Baitarani River basin this climate change assessment using data from three GCM's and considering bias-correction provides new insights. And we do believe that the manuscript is especially relevant for climate change studies focusing on change in rainfall under similar data limited conditions that occur in many parts of the world.*

"Some of the important citations like Chaturvedi et al., 2012; Thrasher et al., 2012; Rana et al., 2014 are missing in the reference list."

> *Due to this omission in the reference list, we checked the list thoroughly. The references to Chaturvedi et al. (2012); Thrasher et al. (2012), and Rana et al. (2014) will be added to the reference list. Additionally, we found 4 references in the list not mentioned in the main text: Guhathakurta (2007), Mitchell and Jones (2005), Mitra and Mishra (2014), and Patwardhan et al (2014). We exclude these references from the final reference list.*

---

## Author Comment (AC2)

**Reply to review comments Referee #1**

*We would like to thank the reviewer for her/his extensive review of our manuscript in the HESS-category 'Cutting-edge case studies' and the sound topics brought forward. The review comments on the bias correction approaches helped us to clarify our reasoning. In this reply we explain why data scarcity even determined the bias correction approaches that could be selected. We also want to emphasize that as hydrologists we are forced to use what data is available in an applied research project. The manuscript benefits from the 'specific comments' on some of the manuscript text and figures. In the following sections we reply (in italics) to all of the reviewers comments (cited):*

**General comments**
"This study explores changes in precipitation over the Brahmani-Baitarani river basin (51 822 km2) in India using three GCMs runs from CMIP5 and two basic bias-correction methods (delta change and linear scaling). Two key challenges that the study is addressing are the low station density and the lack of dynamically downscaled climate simulations. This paper provides interesting insights into challenges faced in regions of low data availability in the context of impact modeling. It could formulate general recommendations on the selection of bias-correction methods in those regions, but to achieve this goal, I consider that further research and major modifications are necessary."

> *The reviewer correctly summarizes our first objective - indeed we try to make a best possible climate change assessment in the absence of dynamically downscaled climate simulations for a basin where we have very limited open observed data at hand. With this manuscript we want to demonstrate the challenges one is faced with in these kind of situations – and we present the best possible effort to derive climate change projections in the absence of reliable reference data.*

"The authors rule out bias-correction methods more complex than delta change and linear scaling because of low data availability. The fundamental question this choice poses is: "What are the data requirements for different bias-correction methods, and in particular, how long should time series be?". I argue that this question should be carefully addressed by the authors on the basis of i) the existing literature (e.g., Rajczak et al. (2015) and references therein), ii) general statistical considerations (e.g., how many data points are required to constrain the distribution of the tails, which can then be used for bias-correction) and possibly, iii) by conducting further numerical experiments (e.g. illustrate the risks of applying complex bias-correction methods with insufficient data by highlighting suspicious/unrealistic features in postprocessed time series)."

> *Most peer-reviewed papers on advanced down-scaling and bias-correction methods address the ideal situation where long historic observational records are available, yet in reality these data may not be present. As the reviewer suggested we have studied the recent paper of Rajczak et al. (2015). We do appreciate the idea of spatially transferring bias-correction functions. However, Rajczak et al. already mention the limitations for spatial heterogeneous variables such as precipitation especially when the reference datasets are short. We have only been provided with the data of a limited number of gauges. We compared the rainfall CDF's of the CWC gauges for the Mountainous and Delta region and in general they tend to overlap, which would allow for the spatial transfer of bias-correction functions from one gauge to another. Yet, the scarce density (more specific: 11 gauges with complete records in a basin of 51 822 $km^2$) does not result in a sufficient spatial grid coverage.*

> *In addition, the derivation of representative transfer functions is ideally based on long-term data records, this to include the natural variability sufficiently with anomalous wet and dry periods. Rajczak et al. (2015) suggest the use of a climatological representative period of at least 20 years. Within their study they show that by using only a limited number of years for the calibration of a quantile mapping transfer function large biases in absolute monthly precipitation amounts can be obtained. For all CWC gauges we have the same time period which does not overlap enough with the GCM reference period (GCM = 1961-1990, CWC = 1990-2012 but only used until 2007 due to overlap with APHRODITE).*

*This confirms the statements we made that an advanced method such as quantile mapping cannot be applied in our situation, with the gauge observations we have at hand as reference, since we do not have rain gauge observed data and GCM data that overlap in time and the observational records only have a limited length. Differences in duration curves may as well occur due to trends in precipitation or incomplete inclusion of natural variability.*

*In addition, the quantile mapping method requires reliable CDFs of reference rainfall and from the comparison between the APHRODITE dataset and station observations (Rana et al., 2014) we know that large biases in rainfall extremes exist indicating that the APHRODITE dataset can neither be used as reference for quantile mapping*

*We add the following sentence to the manuscript (P8L3):*

> "For the derivation of representative transfer functions for the Quantile Mapping approach the calibration is based on long-term data records, ideally more than 20 years, this to include the natural variability sufficiently with anomalous wet and dry periods (Rajczak et al., 2015). Using only a limited number of (overlapping) years, which would be the case if we use the CWC gauge data as reference, can result in large biases in absolute monthly rainfall amounts. When working with non-overlapping periods, an additional difference can be introduced by trends in rainfall time-series and according to Goswami et al. (2012) there has been an increasing trend in both frequency and intensity of heavy rain events over Central India of 10% per decade since 1950.
>
> Moreover the Quantile Mapping method requires reliable CDFs of reference rainfall and from the comparison between the APHRODITE dataset and station observations (Rana et al., 2014) we know that large biases in rainfall extremes exist indicating that the APHRODITE dataset cannot be used as reference for Quantile Mapping."

"Overall, the way the authors deal with extreme events lacks consistency. On one hand, they exclude a large fraction of the available bias-correction methods because of low data availability (P8L6-10). On the other hand, they use the same observations to fit a Gumbel distribution and estimate rainfall intensities associated with return periods of up to 100 years (P10L10, I suggest by the way that the authors provide uncertainty estimates of the intensity of these rare events). If this can be done, why not use this information to bias-correct GCM simulations?"

> *Within this case-study, and similarly for flood risk assessments in general, the stakeholder also is interested in precipitation intensities occurring at high return periods. In the end - to simulate likely flood extents - precipitation intensities occurring at return periods of 50 to 100 years are required input and therefore we derived these Gumbel distributions. We agree with the reviewer that, using the described methods, extremes and changes therein cannot reliably be assessed and estimates are highly uncertain. Yet, as discussed above, the available datasets do not allow for the application of more advanced methods. This is again one of the limitations one is faced with during climate change assessments in countries where open data availability is low.*

> *We added a sentence on the above and the uncertainties in the Gumbel distributions to the manuscript. See the Italic text in the following text block taken from the manuscript:*

> The return period analysis once more illustrates the importance of focussing on relative changes and *'the large uncertainty one will find for precipitation extremes when applying these simple bias-correction methods.'* The precipitation amounts for a 100 year return period of a one-day rainfall event for the period 1990–2013 are 378, 328, and 229 mm for Akhuapada,

Anandpur and Tilga rainfall gauge respectively, whereas the climatological data and 20 the baseline rainfall for the LS method applied to all GCMs and regions show intensities of 100–200 mm for the same return period. Therefore, translating absolute future climate results of extreme rainfall events based on the climatological dataset to rainfall intensities recognizable for local authorities familiar with the rainfall gauges remains a challenge', *yet these are required statistics for flood risk assessments'(P16L16-24).*

*In addition the Gumbel distributions for the reference situation / observations are based on the APHRODITE dataset. This was not clear in the original manuscript and we adjust this in our revision. Thus the data used as reference / observations in the Gumbel distributions has been used for the bias-correction of the GCM data.*

"Similarly, the authors acknowledge that the postprocessing methods they selected "only concentrate on the correction of monthly mean rainfall amounts and can thus affect our analysis of changes in extreme rainfall indices". I certainly understand that they are limited by the data, but I urge them to critically assess and discuss how much can be learned about extreme events when methods such as delta change and linear scaling are used."

*The drawback of the Delta Change method is clear – changes in monthly mean precipitation amounts are projected on all daily rainfall amounts whereas we know that precipitation extremes are likely to increase more (Goswami et al. 2012 and others). As a result we will underestimate future precipitation extremes – this can also be seen in Figure 11 where future extremes obtained with the DC method are lower than future extremes obtained with the LS method.*

*For the linear scaling approach it is less clear what will happen. The full distribution will change following in general the changes of the uncorrected GCM time-series. The main question is whether the scaling of the reference GCM time-series increases its similarity with the observed data. We visualized the difference by plotting CDFs of the corrected and un-corrected GCM reference time-series together with the APHRODITE CDF. In Figure A below, which only show the extreme end of the CDFs, this would mean that the red line should be closer to the green than the black. For HadGEM2-ES we conclude that this is the case for both regions. For MIROC-ESM and GFDL-CM3 we have reached this goal at least for the Mountainous region. For the latter two GCMs we do not find a pronounced improvement for the Delta region and projected changes in extremes will therefore also be less reliable here. The large bias and limited improvement is likely a result of the location of the GCM cell that encompasses both sea and land.*

[Figure]

*Figure A. CDF curves of daily rainfall for the period 1960-1990 for the GCM uncorrected (black line), GCM LS-corrected (red line) and APHRODITE (green line) data set for the monsoon season.*

"It is really a matter of communicating meaningful information, especially when the results are then used for decision making. One way to overcome the lack of data is to make stronger links to processes leading to floods (see the related discussion on the consideration of misrepresented process when bias-correcting models in Addor et al., 2016). It means identifying those processes (e.g. storm surges described on P10L16-23 or specific aspects of the monsoon), evaluate how they are captured by the models, and then, instead of simply looking at how precipitation is projected to change, look at how these processes are projected to change. Their scale is probably large enough to be captured by GCMs. If this link is made, then the low station density in the region would be less problematic and the study could be a major breakthrough."

*For the selection of GCMs we rely on the work of Menon et al. (2013) and Chaturvedi et al. (2012) where GCM performance is evaluated for seasonal (JJAS) mean rainfall (Menon et al., 2013) and annual mean rainfall (Chaturvedi et al., 2012).*

*The above suggestion of the reviewer is very valuable and could indeed improve bias-corrections and the evaluation thereof in local climate change assessments in areas with limited data availability. This could be done by focussing GCM evaluation on the representation of large scale circulation processes with for example simulations from historical numerical weather re-analysis as reference as is described in Addor et al. (2016). Yet, such an analysis requires a thorough understanding and analysis of synoptic circulation patterns which is of great value for an advanced research paper on this topic but behind the scope of an applied impact assessment. Moreover, although from the point of atmospherical processes more correct, for us the conservation of inter-variable relations is less relevant as we focus our analysis on a single variable.*

*We add a comment on the potential value of the analysis and conservation of synoptic circulations to the discussion of our manuscript (P18L14):*

"Future assessments could possibly be improved by applying the method proposed by Addor et al. (2015). They suggest to evaluate how the underlying circulation processes are projected to change and that from there the potential changes in precipitation patterns, means and extremes could be assessed."

**Specific comments**

"The authors "focus on three models that nearly span the full range of annual mean rainfall projections available for India" (P4L19-20). Please also comment on how well they cover the range of projected changes in extreme events, which are also a main focus of the study."

> *Menon et al. (2013) focuses on GCM performance for seasonal mean (JJAS) rainfall and unfortunately no results on how well they cover extreme events are described. We adjusted 'annual' in 'seasonal' in P4L19-20 of our manuscript.*

"P5L18-19: Why are the authors using 30 years for the reference period and then 40 years for the future conditions? This should be harmonized."

> *We agree with the reviewer that these types of analyses must be harmonized. The reference period is 1961-1990, and the future conditions span 2030-2059 and 2070-2099. The second period was wrongly stated in the manuscript. Thus, our analysis is therefore harmonized, and we change the corresponding years from 2040 and 2080 into 2045 and 2085 respectively.*

"Section 2.3: Please briefly explain how orographic effects were accounted for when generating the APHRODITE data set."

> *Yatagai et al. (2012) applied an angular distance weighting. Quote:*
> *"A small weight is given to target cells on the leeward of a high ridge if the ridge was between the target cell and its nearest rain gauge. A large weight is given to a target cell on a slope that inclines to a rain gauge. A lookup table is constructed for each month to define the correlation distance […] and used to define the weighting function."*

"Were measurements from the three CWC rain gauges also used to produce the APHRODITE data set? "

> *Tabel 1 in Yatagai et al. (2012) shows that the Indian Meteorological Department provided daily rainfall data. No reference is made to the use of CWC rain gauge data. We contacted Prof. Yatagai to clarify this. Unfortunately, we did not get a reply to date.*

"Which period is covered by the three gauges?"

> *The CWC rain gauges cover the 1990-2012 period but only used until 2007 due to overlap with APHRODITE.*

"The authors write that "gauge observations could not be used as reference since they do not overlap with the full GCM baseline period" (P7L6-7), please develop. "

> *We kindly refer to our reply to the comment on data requirements of different bias correction approaches.*

"P12L5-15: The differences reported between the two observational datasets are indeed large, it is in particular clear from Figure 4, although it is not completely surprising since the CWC curves are produced using solely three stations for 51 822 km2. Yet I think the observational uncertainty should be better accounted for and displayed, as it influences the bias-correction and thereby the projected changes."

> *Each CDF has been produced from the individual time-series of one CWC gauge and the corresponding grid cell. One CDF thus only presents information from one observational record. We wanted to evaluate the comparability of APHRODITE and CWC gauge data focussing on the CWC locations for which the time-series are familiar to the Indian water managers. We do not have any uncertainty information for these single points – and therefore only focus on the absolute differences.*

"Also, I understand that the "climatological data set" is APHRODITE, if it is the case, I would state it more clearly."

*We agree that the manuscript benefits from stating this more clearly. We change 'climatological dataset' to 'APHRODITE' throughout the manuscript. We also change P7L2-3 in the method section of the manuscript into the following comment.*

"Hereafter it is applied as the climatological rainfall dataset and referred to as APHRODITE."

"P5L20-21: "Almost all CMIP5 models show good performance for surface air temperature simulations averaged over South Asia and the Indian Sub-continent", please add a reference."

*We add reference to Chaturvedi et al. (2012) in the manuscript.*

"P4L9: need"

*We agree with the reviewers' linguistic comment and change it in the manuscript.*

"[…] and need to be corrected to ensure its applicability at the local scale"

"P6L11-12: I suggest using 'wide' instead of 'full' "

*We agree with the reviewers' suggestion and change this accordingly.*

"[…] and span a wide range of socio-economic driving forces"

"P6L19: What is Monsoon Asia?"

*Yatagai et al. (2012) uses this term in their APHRODITE paper to describe the domains assessed. Figure 1 shows the region Yatagai et al. (2012) refer to as Monsoon Asia. We include Figure 1 in this reply.*

[Figure]

FIG. 1. The domains and rain gauge distributions used in APHRODITE V1101 for monsoon Asia (MA), the Middle East (ME), and northern Eurasia [i.e., Russia (RU)], and in V1005 for Japan (JP). Stations derived from the GTS network (blue dots), those from the precompiled dataset (black dots), and those in APHRODITE's individual data collection (red dots).

"P7L20: Out of curiosity: were RCMs ever run over India, for instance under SRES emission scenarios, and if yes, please provide references and discuss what this revealed."

*A rather simplified regional climate modeling system, known as PRECIS (Providing Regional Climates for Impacts Studies) developed by the Hadley Centre for Climate Prediction and Research, was applied for India to develop high-resolution climate change scenarios. Under this program, Indian Institute of Tropical Meteorology (IITM) generated daily time series of four emission scenarios (i.e. CMIP3 scenarios A1, A2, B1, and B2) at spatial resolution of 0.44°X 0.44° (Rupa Kumar et al. 2006). The RCM was able to resolve features on finer scales than those resolved by the GCM, particularly those related to improved resolution of the topography. The most notable advantage of using the RCM is a more realistic representation of the spatial patterns of summer monsoon rainfall such as the maximum along the windward side of the Western Ghats of India (Lucas-Picher et. al., 2011).*

*The mean annual cycles of all-India mean precipitation and surface air temperatures for A2 and B2 scenarios derived with the RCM indicate a general increase in precipitation and temperature, for the country as a whole. Spatial patterns of rainfall change indicate maximum increase over west coast and northeast India for both A2 and B2 scenarios. PRECIS estimates 20% rise in all India summer monsoon rainfall in future scenarios as compared to present. In the future scenarios PRECIS simulation for 2071–2100 indicates an all-round warming over Indian subcontinent associated with increasing greenhouse gas concentrations.*

"P13L14: 900mm/month?"

*It is 900 mm/day. HadGEM2-ES and MIROC-ESM underestimate the monsoon rainfall in June. The LS method consequently leads to a large correction factor, see Table 6. As storm events do occur in June, these large correction factors, ranging from 4.6-11.1, 'explodes' a rather heavy rainfall event (35.6-64.4mm) or a heavy rain event (64.4-124.4mm) to an almost unrealistic extreme rainfall event. We checked this with recorded values in India. Guhathakurta (2007) provides an overview of record rainfall for different durations over India. The maximum recorded one day (24hrs) rainfall is 1563mm. Although up to 900mm day-1 could therefore be a realistic value in India, it is not in line with CWC observations over the 1990-2007 period. This data shows a maximum one day rainfall of 396.8mm. In the assessed periods, climate change will most likely not double (x2.2) the maximum daily rainfall in June. We therefore applied the July multiplier to June as described in section 3.3.*

"General comments on the figures: I encourage the authors to use colors, that is free of charge when publishing in HESS, and it would make their figures easier and more enjoyable to read. I recommend using full names in the captions instead of acronyms (e.g. MRD, HRD) and using a larger font size. "

*We agree with the reviewers' encouragement to use colors and full names in the captions. Our initial reasoning was we want our manuscript to be readable for experts and agencies in developing countries which sometimes lack easy access to color printers. These colors can be viewed on screen and we add an example of the updated figures.*

[Figure]

"**Figure 7.** Difference in number of Wet Days: 2045 short term climate projection (dark blue: LS, dashed dark green: DC), and 2085 long term climate projection (light blue: LS, dashed light green: DC)."

"Figure 11: this figure definitively needs uncertainty bounds."

*We agree with the reviewer that the figure benefits from adding uncertainty bounds. We adjusted the figure accordingly.*

[Figure]

"**Figure 11**. Return periods based on annual maximum. APHRODITE (black), 2045 short term climate projection (dashed blue: LS, dashed green with circle: DC), and 2085 long term climate projection (dotted blue: LS, dotted green with circle: DC)."

Suggested references:

Addor, N., Rohrer, M., Furrer, R. and Seibert, J.: Propagation of biases in climate models from the synoptic to the regional scale: Implications for bias adjustment, J. Geophys. Res. Atmos., doi:10.1002/2015JD024040, 2016.

*We value these suggestions, and we add them to the manuscript. The suggested references have been used in this reply to the reviewer as well as in the proposed new text for the revised manuscript.*

Rajczak, J., Kotlarski, S., Salzmann, N. and Schär, C.: Robust climate scenarios for sites with sparse observations: a two-step bias correction approach, Int. J. Climatol., doi:10.1002/joc.4417, 2015.

*The suggested references have been used in this reply to the reviewer as well as in the proposed new text for the revised manuscript.*